# Role of Arginase-II in Podocyte Injury under Hypoxic Conditions

**DOI:** 10.3390/biom12091213

**Published:** 2022-08-31

**Authors:** Zhilong Ren, Duilio Michele Potenza, Yiqiong Ma, Guillaume Ajalbert, David Hoogewijs, Xiu-Fen Ming, Zhihong Yang

**Affiliations:** 1Cardiovascular & Aging Research, Department of Endocrinology, Metabolism, Cardiovascular System, Faculty of Science and Medicine, University of Fribourg, CH-1700 Fribourg, Switzerland; 2Department of Nephrology, Renmin Hospital of Wuhan University, Wuhan 430060, China; 3Integrative Oxygen Physiology, Department of Endocrinology, Metabolism, Cardiovascular System, Faculty of Science and Medicine, University of Fribourg, CH-1700 Fribourg, Switzerland

**Keywords:** arginase, hypoxia, HIF, podocytes, ROS

## Abstract

Hypoxia plays a crucial role in acute and chronic renal injury, which is attributable to renal tubular and glomerular cell damage. Some studies provide evidence that hypoxia-dependent upregulation of the mitochondrial enzyme arginase type-II (Arg-II) in tubular cells promotes renal tubular injury. It is, however, not known whether Arg-II is also expressed in glomerular cells, particularly podocytes under hypoxic conditions, contributing to hypoxia-induced podocyte injury. The effects of hypoxia on human podocyte cells (AB8/13) in cultures and on isolated kidneys from wild-type (*wt*) and *arg-ii* gene-deficient (*arg-ii^−/−^*) mice ex vivo, as well as on mice of the two genotypes in vivo, were investigated, respectively. We found that the Arg-II levels were enhanced in cultured podocytes in a time-dependent manner over 48 h, which was dependent on the stabilization of hypoxia-inducible factor 1α (HIF1α). Moreover, a hypoxia-induced derangement of cellular actin cytoskeletal fibers, a decrease in podocin, and an increase in mitochondrial ROS (mtROS) generation—as measured by MitoSOX—were inhibited by adenoviral-mediated *arg-ii* gene silencing. These effects of hypoxia on podocyte injury were mimicked by the HIFα stabilizing drug DMOG, which inhibits prolyl hydroxylases (PHD), the enzymes involved in HIFα degradation. The silencing of *arg-ii* prevented the detrimental effects of DMOG on podocytes. Furthermore, the inhibition of mtROS generation by rotenone—the inhibitor of respiration chain complex-I—recapitulated the protective effects of *arg-ii* silencing on podocytes under hypoxic conditions. Moreover, the ex vivo experiments with isolated kidney tissues and the in vivo experiments with mice exposed to hypoxic conditions showed increased Arg-II levels in podocytes and decreased podocyte markers regarding synaptopodin in *wt* mice but not in *arg-ii^−/−^* mice. While age-associated albuminuria was reduced in the *arg-ii^−/−^* mice, the hypoxia-induced increase in albuminuria was, however, not significantly affected in the *arg-ii^−/−^*. Our study demonstrates that Arg-II in podocytes promotes cell injury. *Arg-ii* ablation seems insufficient to protect mice in vivo against a hypoxia-induced increase in albuminuria, but it does reduce albuminuria in aging.

## 1. Introduction

Glomerular podocytes are highly differentiated cells that wrap around the renal glomerular capillaries and are integral cellular members of the glomerular filtration barrier [1]. The organized microtubular and actin cytoskeletons in podocytes are important for the formation of their foot processes and slit diaphragm to cover the basement membrane and exert the normal function of glomerular filtration [2]. Because podocytes have only a limited capability to renew, damage to the cells, for example, by hypoxia or dysfunction associated with old age, leads to irreversible glomerular dysfunction and chronic kidney disease [2,3].

Experimental research provides a body of data demonstrating that hypoxia, a deficiency of oxygen supply, is prevalent during renal injury, owing to the high metabolic demand for kidney function [4]. Oxygen is essential for aerobic respiration to generate ATP as energy fuel for the ATP-dependent transporters in the kidney so that basic renal functions, i.e., the balance of the secretion and absorption of water, electrolytes, and nutrient as well as the acid-base balance, can be maintained [5]. It is well-evidenced that acute and chronic hypoxia play an important role in the development of renal failure [6,7]. Some studies demonstrate that hypoxia causes injury to various nephron cell types, including podocytes, resulting in slit-diaphragm dysfunction, foot-process effacement, and cytoskeleton derangement, which are linked to an accumulation of hypoxia-inducible factors (HIFs) [8,9,10,11]. HIFs are the master mediators of the cellular transcriptional response to hypoxia and belong to the basic-helix-loop-helix PER-ARNT-SIM (bHLH PAS) domain protein superfamily [12]. HIFs are heterodimeric transcription factors consisting of an oxygen-regulated α-subunit, including HIF1α, HIF2α and HIF-3α and a constitutively expressed β-subunit (HIFβ) [13]. The most well-studied HIFs are HIF1α and HIF2α, which form heterodimers with HIFβ as HIF1 and HIF2 complexes, binding to the hypoxia-responsive elements (HREs) on promoters and enhancers of various target genes, and activates transcription [12,14]. Under hypoxic conditions, HIF1α and HIF2α escape hydroxylation by prolyl-hydroxylase (PHD), so that recognition and ubiquitination by von Hippel-Lindau (VHL) and rapid degradation through the E3 ubiquitin ligase complex do not occur, allowing stabilization of HIFs [15]. Some studies report that the upregulation of HIF1α under hypoxic conditions causes podocyte damage [16]. However, the underlying mechanisms of HIF-induced podocyte damage are not fully understood.

Arginase is an enzyme that metabolizes L-arginine to L-ornithine and urea [17]. Two isoforms of arginase, Arg-I and Arg-II, are present in mammals, and they are encoded by different genes [18]. Arg-I is mainly expressed in hepatocytes, whereas Arg-II is highly expressed in the kidney [19]. Arg-I is located in the cytosol, and Arg-II is expressed in the mitochondrion and is inducible in many tissues/cells [20,21,22]. Studies demonstrate that Arg-II plays an important role in organ damage under various pathological conditions when it is upregulated [20]. Many factors and mechanisms are reported to be responsible for the upregulation of Arg-II [20]. Among them, hypoxia has been proven to be a strong inducer of Arg-II in many cell types due to the fact that the promoter, as well as an enhancer region in *arg-ii* intron 2, contain HIF-binding sites, as revealed by the ChIP-sequencing experiments [23]. In the kidney, Arg-II is mainly expressed in the S3 segment of the proximal tubular cells (PTCs), and our recent study shows that the upregulation of Arg-II in the PTCs contributes to the kidney aging process and renal damage caused by hypoxia [19,24]. Initial studies suggest that Arg-II is exclusively expressed in S3 PTCs [25]. However, recent studies demonstrate that Arg-II is also inducible in other cell types of the kidney, e.g., in collecting ducts under water deprivation, which is involved in renal water balance [26]. A recent study by Li and colleagues [27] demonstrates that Arg-II is inducible in podocytes and plays an important role in type 2 diabetic nephropathy via mitochondrial dysfunction. It is, however, not known whether Arg-II is inducible in podocytes under hypoxic conditions and whether it plays a role in podocyte injury under this condition.

The aim of our study is, therefore, to investigate a possible role of Arg-II in hypoxia-induced podocyte injury and whether this is related to mitochondrial dysfunction.

## 2. Materials and Methods

### 2.1. Reagents

Reagents were purchased or obtained from the following sources: rabbit antibody against Arg-II (#55003) was from Cell Signaling Technology (Danvers, MA, USA); mouse antibody against HIF1α (610958) was from BD Transduction Laboratories (Franklin Lakes, NJ, USA); rabbit antibody against HIF2α (PAB12124) was from Abnova (Taipei, Taiwan); rabbit antibody against podocin (PA5-37284) was from Invitrogen/Thermo Fisher Scientific (Waltham, MA, USA); mouse antibody against synaptopodin (sc-515842) and mouse antibody against ACE1 (sc-23908) were from Santa Cruz Biotechnology (Dallas, TX, USA); mouse antibody against tubulin (T5168), mouse antibody against β-actin (A5441), dimethyloxaloylglycine (DMOG), and rotenone were from Sigma-Aldrich (St. Louis, MO, USA); IRDye 800-conjugated affinity purified goat anti-rabbit IgG F(c) was purchased from LI-COR Biosciences (Lincoln, NE, USA); goat anti-mouse IgG (H + L) secondary antibody Alexa Fluor^®^ 680 conjugate was from Invitrogen/Thermo Fisher Scientific (Waltham, MA, USA).

### 2.2. Generation of Recombinant Adenovirus (rAd)

The rAd-expressing shRNA targeting human HIF1α and HIF2α, driven by the U6 promoter (rAd/U6- HIF-1α/2α^shRNA^), rAd/U6-LacZ^shRNA^ and rAd/U6-Arg-II^shRNA^ were generated and characterized as previously described [24,28,29].

### 2.3. Podocyte Cell Culture

The conditionally immortalized human podocytes (AB8/13) were kindly provided by Andreas Kistler (University of Zürich) [30] and cultured in an RPMI-1640 medium (BioConcept) containing 10% fetal bovine serum (FBS), 100 U/mL penicillin, 100 μg/mL streptomycin (Gibco), 1×insulin, transferrin and selenium (ITS) in a 33 °C incubator with 5% CO_2_ for proliferation. When the podocytes were about 50% confluent, the cells were moved from 33 °C to 37 °C for differentiation for 10–14 days. For the staining experiments, AB8/13 were cultured on coverslips coated with 1% gelatin. Before all the experiments, the differentiated podocytes were serum starved in the RPMI-1640 medium without FBS overnight. Then, the cells were cultured in a Hypoxic Cabinet System for the in vitro studies (1% O_2_, The Coy Laboratory Products, Grass Lake, MI, USA) or treated with 1 mmol/L DMOG (inhibitor of prolyl-hydroxylase, which was involved in HIFα degradation) for different times (6 to 72 h). For the adenoviral transduction experiments, AB8/13 were transduced with the rAd at titers of 200 multiplicities of infection (MOI) and then cultured in a complete medium for two days and then switched to a serum-free medium overnight before the experiments.

### 2.4. Generation of Arg-ii Knockout Cell Line Using CRISPR/Cas9 Technologies

The sgRNA targeting human *arg-ii* (the top strand of the sgRNA that recognizes the target DNA region of human *arg-ii*: GGGACTAACCTATCGAGA) was cloned into pSpCas9(BB)-2A-Puro (PX459) V2.0 (Plasmid #62988, Addgene) to generate pSpCas9(BB)-2A-Puro (PX459)-U6/sgRNA-harg-ii. The AB8/13 cells were plated in a 6-cm dish at a density of 1 × 10^6^ cells 24 h before transfection. The transfection of pSpCas9(BB)-2A-Puro (PX459)-U6/sgRNA-*arg-ii* was performed using the Lipofectamine™ 3000 Transfection Reagent (L3000008, Invitrogen™) according to the manufacturer’s protocol. Briefly, per 1 × 10^6^ cells, diluted plasmid DNA (5 µg, diluted with P3000™ Reagent) and diluted Lipofectamine™ 3000 Transfection Reagent were mixed at a 1:1 ratio and incubated at room temperature for 15 min. The DNA-lipid complex was then added to the cells. To select the sgRNA-positive cells 48 h post-transfection, the cells were treated with puromycin (2.5 µg/mL) for 48 h until all the control cells without transfection died. The puromycin-resistant cells were allowed to recover in a medium without puromycin for 1 week before seeding the single cells into 96-well plates by using the dilution method. Single clones were then expanded and screened for Arg-II by immunoblotting. Arg-ii knockout was then confirmed by immunoblotting.

### 2.5. Immunoblotting

The cells were lysed on ice with lysis buffer containing 20 mmol/L Tris.HCl, 138 mmol/L NaCl, 2.7 mmol/L KCl, pH 8.0, supplemented with 5% glycerol, 1 mmol/L MgCl_2_, 1 mmol/L CaCl_2_, 1 mmol/L sodium-o-vanadate, 20 μmol/L leupeptin, 18 μmol/L pepstatin, 1% NP-40, 5 mmol/L EDTA, and 20 mmol/L NaF. After being frozen and thawed twice, repeatedly with liquid nitrogen, the cell lysates were centrifuged at 12,000 rpm for 20 min at 4 °C. The supernatants were taken, and the protein concentration was determined using the Lowry protein assay. Equal amounts (30 μg) of protein were mixed with a loading buffer and boiled at 95 °C for 5min, and separated using 10% sodium dodecyl sulfate-polyacrylamide gel electrophoresis (SDS-PAGE) before being transferred to an Immobilon-P membrane (Millipore), and the resultant membrane was incubated overnight with the corresponding primary antibody at 4 °C with gentle shaking after being blocked with 5% skimmed milk. The blot was then further incubated with a corresponding anti-mouse (Alexa Fluor 680 conjugated) or anti-rabbit (IRDye 800-conjugated) secondary antibody. The signals were visualized using the Odyssey Infrared Imaging System (LI-COR Biosciences, Lincoln, Nebraska, USA). Quantification of the signals was performed using the NIH Image J 1.50i software (National Institutes of Health, Bethesda, MD, USA). The information on the antibodies used for immunoblotting is presented in Appendix A.

### 2.6. Cytoskeleton Staining

The cells grown on coverslips were fixed with 4% formaldehyde at room temperature for 30 min and washed 3 times with PBS for 5 min. Then, the fixed cells were incubated with 0.4 nmol/mL Atto 488 phalloidin (Sigma-Aldrich, (St. Louis, MO, USA) for F-actin staining for 60 min at room temperature, and the DNA staining dye 300 nmol/L DAPI (4′6-diamidino-2-phenyl-indole dihydrochloride, Invitrogen) was added simultaneously. Images were acquired through 40× objectives with a Leica TCS SP5 confocal laser microscope. Ten fields of view were randomly selected, and the percentage of podocytes with disrupted cytoskeleton actin was quantified.

### 2.7. Mitochondrial ROS

An amount of 1 mL of 5 μmol/L MitoSOX^TM^ reagent working solution (Molecular Probes^TM^ Invitrogen detection technologies, Waltham, MA, USA) with 1 μg/mL Hoechst 33342 (Thermo Scientific (Waltham, MA, USA)) was applied to cover the cells adhering to the coverslips. The cells were incubated for 10 min at 37 °C, protected from light. Then, the cells were gently washed three times with warm Krebs buffer. Images were acquired through 40× objectives with a Leica TCS SP5 confocal microscope.

### 2.8. Animal Experiments

The *Arg-ii*^−/−^ mice were kindly provided by Dr William O’Brien [31] and backcrossed to C57BL/6J for more than 10 generations. The genotypes of the mice were confirmed by a polymerase chain reaction (PCR) as previously described [31]. The offspring of the *wt* and *arg-ii*^−/−^ mice were generated by interbred from the hetero/hetero cross. Mice were maintained on a 12:12 h light/dark cycle at 23 °C with free access to standard laboratory feed and water. All animal protocols were approved by the Ethical Committee of the Veterinary Office of Fribourg Switzerland (2020-01-FR) and were performed in compliance with the guidelines on animal experimentation at our institution.

### 2.9. Ex Vivo Experiments with Isolated Kidneys

The *wt* and *arg-ii*^−/−^ male mice, at the age of 22 months, were anesthetized and sacrificed by exsanguination. The kidneys from both the *wt* and *arg-ii*^−/−^ mice were quickly excised, sliced horizontally, and immersed in RPMI-1640 medium into a 6-well plate. The media were supplemented with insulin-transferrin-selenium (ITS) and penicillin/streptomycin (1%). The tissues were incubated in a Coy In Vitro Hypoxic Cabinet System (The Coy Laboratory Products, Grass Lake, MI, USA) at 1% O_2_ with a premixed gas of 5% CO_2_/95% N_2_ or in normoxic chambers as controls for 24 h. Then, the kidney tissues were fixed with 3.7% paraformaldehyde and then embedded in paraffin for the immunofluorescence staining experiments.

### 2.10. In Vivo Hypoxia Experiments with Animals

The *wt* and *arg-ii*^−/−^ male mice, at the age of 22 to 24 (n = 13 for genotypes), were randomized into two groups that were exposed to the normoxia or hypoxia conditions. The mice of the hypoxia group were transferred within their home cages into a Hypoxic System for the in vivo experiments (Coy Laboratory Products Inc. Grass Lake, MI, USA), where a normobaric hypoxia of 8% oxygen was achieved by mixing oxygen with N_2_. After an adaptation period at normoxic conditions (21% O_2_), the oxygen level was gradually lowered at an increment of 2% every 20 min. The final 8% normobaric hypoxia was achieved after 2 h. The animals were exposed to hypoxia for 24 h, during which they were monitored at least 2 times to verify their general health condition. At the end of the experiment, the animals were anesthetized, euthanized, and the kidneys removed. The kidney samples were either used for immunoblotting or for the immunofluorescence staining experiments.

### 2.11. Measurement of Urinary Creatinine and Albuminuria

Urine was collected on hydrophobic sand (LabSand^®^, Coastline Global, CA, USA) according to the manufacturer’s instructions [32]. The animals were food deprived of access to water for the duration of the urine collection (12 h). Mice were placed alone in solid bottom cages with an adequate quantity of LabSand^®^, ~0.5 cm, to cover the bottom of the cage. Urine drops were collected each hour with a pipette and transferred to a polypropylene tube with a cap. Urine creatinine was then measured by a colorimetric assay using a creatinine urinary detection kit (EIACUN, Invitrogen/Thermo Fisher Scientific, Waltham, MA, USA). Briefly, the urine was diluted at 1:25 with distilled water. The 50 µL diluted samples and standards were incubated with 100 µL Creatinine Reagent for 30 min. The absorbance was measured at 490 nm. Albuminuria was measured by ELISA using a mouse Albuwell kit (1011, Exocell Inc., Philadelphia, PA, USA), according to the manufacturer’s instructions [33]. Briefly, the urine samples were diluted at 1:25 with NHEBSA, and 50 µL of the diluted samples and standards were incubated with 30 µL of anti-mouse Albumin Ab-HRP conjugate for 30 min. Subsequently, 100 µL of Color Developer was added to each well and incubated for 10 min. The reaction was stopped by adding 100 µL of Color Stopper. The absorbance was measured at 450 nm. Albumin was normalized by creatinine, and albuminuria was quantified by the albumin/creatinine ratio (uACR, μg/mg). Both creatinine and albumin were measured from 12 h of daytime urine collected from the animals.

### 2.12. Confocal Immunofluorescence Staining of Arg-II, Synaptopodin, and ACE1

Kidneys from the *wt* and *arg-ii^−/−^* mice were isolated and fixed with 3.7% paraformaldehyde and embedded in paraffin. Horizontal central transverse sections through the middle of the kidney (5 μm) were prepared with Microtome. After deparaffinization in xylene (2 times, 10 min for each), hydration in ethanol (twice in 100% ethanol, twice in 95% ethanol, and once in 80%, for 3 min, sequentially), and antigen retrieval in EDTA Buffer (1 mmol/L EDTA, 0.05% Tween 20, pH 8.0) in a pressure cooker, paraffin-embedded sections (5 μm) were first blocked for three hours with mouse Ig-blocking buffer and then with 1% BSA and 10% goat serum in PBS for 60 min and were then incubated with first antibodies (Arg-II 1:100, synaptopodin 1:100, ACE1 1:50) at 4 °C overnight. The following day, the slices were further incubated with Alexa Fluor 488–conjugated goat anti-rabbit IgG (H + L) or goat anti-mouse IgG (H + L) for 2 h at room temperature in darkness, followed by counterstaining with 300 nmol/L DAPI for 5 min. For the co-localization experiments, Alexa Fluor 568–conjugated goat anti-mouse IgG (H + L) and Alexa Fluor 488–conjugated goat anti-rabbit were used to detect synaptopodin and Arg-II, respectively. Negative controls were performed by omitting the primary antibodies. The immunofluorescence signals were visualized using the Leica TCS SP5 confocal microscope. The intensity of the fluorescence was quantified by a densitometric analysis of the target proteins by using the open-source software ImageJ (Fiji). Specifically, the background signal was determined in the non-stained areas in each section. The positive signals were outlined manually according to their signal.

### 2.13. Statistics

Data are given as a mean ± SD, and all the experiments in this present study were performed independently at least three times. The Kolmogorov–Smirnov test was used to determine the Gaussian distributions. Statistical analyses were performed with the Student’s *t*-test for the paired or unpaired observations or ANOVA with Bonferroni’s post-test; *p* < 0.05 was considered statistically significant.

## 3. Results

### 3.1. Hypoxia Enhances Arg-II Levels in Human Podocytes through HIF1α

Exposure of the differentiated human podocyte cells to hypoxia over 48 h enhanced the Arg-II levels in a time-dependent manner, which was accompanied by enhanced levels of HIF1α and HIF2α (Figure 1A–D). No Arg-1 could be detected under this condition. While the elevated HIF1α levels under hypoxia gradually decreased over the time period, the HIF2α levels remained elevated (Figure 1A,C,D). These effects of hypoxia on Arg-II and HIFs were mimicked by our treatment of the cells with the prolyl-hydroxylase (PHD) inhibitor, DMOG (1 mmol/L), which stabilizes HIFα subunits by preventing degradation under normoxic conditions (Appendix A). To investigate the role of the HIF isoforms in the regulation of Arg-II under hypoxic conditions, *hif1α*, *hif2α* or both genes were silenced by rAd-shRNA, respectively (Figure 2A). As expected, hypoxia-induced accumulation of HIF1α was significantly inhibited by silencing *hif1α* but not by silencing *hif2α* (Figure 2A,C), resulting in the inhibition of Arg-II up-regulation under hypoxic conditions (Figure 2A,B,C). Of note, silencing *hif1α* was accompanied by partial inhibition of hypoxia-induced HIF2α accumulation (Figure 2A,D). However, silencing *hif2α* did not affect the Arg-II levels under hypoxic conditions (Figure 2A,B,D), demonstrating that HIF1α (but not HIF2α) mediates Arg-II upregulation under hypoxic conditions in podocytes.

### 3.2. Silencing Arg-II Reduces Hypoxia-Induced Cytoskeleton Filament Derangement

We next investigated the influence of hypoxia on the cytoskeleton filament organization in human podocytes. Differentiated podocytes under normoxic conditions displayed long, parallel, and organized cytoskeleton filament fibers (Appendix A). The cytoskeleton fiber organization pattern became shortened and disrupted under hypoxic conditions, and the percentage of the cells with cytoskeleton fiber disruption increased over 48 h (Appendix A). As confirmed by immunoblotting, silencing *arg-ii* in the cells prevented an increase in the Arg-II levels stimulated under hypoxic conditions (Figure 3A) and reduced hypoxia-evoked cytoskeleton fiber derangement (Figure 3B). Similar to hypoxia, treatment of the cells with the HIF stabilizing agent DMOG caused cytoskeleton fiber derangement, which was significantly reduced by silencing *arg-ii* (Appendix A).

### 3.3. Silencing Arg-II Prevents Hypoxia-Mediated Decrease in Podocin Levels

Podocin has been reported to be closely associated with podocyte cytoskeleton actin organization. Therefore, we further analyzed the role of Arg-II in podocin expression under hypoxic conditions. As shown in Figure 3C,D, hypoxia over 48 h decreased the podocin levels in the podocytes, which was prevented by silencing *arg-ii*, demonstrating a role of Arg-II in hypoxia-induced podocin downregulation.

### 3.4. Silencing Arg-II Prevents Mitochondrial ROS Production

In the podocytes exposed to the hypoxic conditions for 48 h, mitochondrial ROS (mtROS) was elevated, as demonstrated by the MitoSOX signals (Figure 4A,B). The enhanced mtROS was prevented by silencing *arg-ii* (Figure 4A,B). The stabilization of HIFs by the PHD inhibitor, DMOG (1 mmol/L), under normoxic conditions, mimicked the hypoxia-dependent effect on mtROS generation, which was prevented by silencing *arg-ii* in the cells (Appendix A).

### 3.5. Role of mtROS in Hypoxia-Induced Podocyte Injury

As shown in Figure 5A,B, under hypoxic conditions, the increase in the cellular production of mtROS, as demonstrated by the MitoSOX signals examined under an immunofluorescence confocal microscope, was inhibited by the mitochondrial complex-I inhibitor rotenone (2 μmol/L for 1 h). To further explore the role of mtROS in hypoxia-induced podocyte injury, the cells were treated with rotenone (2 μmol/L) for 1 h, as described above. As shown in Figure 5C,D, the increase in the percentage of cells with cytoskeleton fiber derangement under the hypoxic conditions was reduced by rotenone treatment, while in the cells deficient in *arg-ii* (*arg-ii^−/−^*, Appendix A), neither hypoxia nor rotenone had a significant effect on the cytoskeleton fiber alterations (Appendix A), demonstrating that Arg-II exerts its effect through mtROS. Synaptopodin, an important marker of podocyte function, was decreased in the hypoxic conditions, which was prevented by rotenone (Figure 5E,F). The results demonstrate an important role of mtROS in hypoxia-induced podocyte injury.

### 3.6. Hypoxia Causes Podocyte Injury in Mouse Kidneys Ex Vivo and In Vivo: Prevention by Arg-II Deficiency

Ex vivo experiments with isolated kidney tissues exposed to hypoxic conditions (1% O_2_) for 24 h were performed. This experimental approach avoids the effects of systemic knockout of the *arg-ii* gene on kidneys. The confocal microscopy results revealed that Arg-II in the glomeruli of *wt* mice exposed to hypoxia was enhanced as compared to the normoxic conditions (Figure 6A). Synaptopodin levels were decreased under hypoxic conditions, which was significantly reversed in the *arg-ii^−/−^* mouse kidneys (Figure 6B). Further experiments with the co-staining of synaptopodin and Arg-II revealed a partial co-localization (37%) of the two molecules in the glomeruli of *wt* mice under hypoxic conditions, while 63% of Arg-II are not co-localized with synaptopodin (Figure 6C), demonstrating that Arg-II is induced under hypoxic conditions in podocytes and also in other cells, i.e., mesangial cells and/or endothelial cells. Further experiments with confocal Z-stack recording confirm the localization of Arg-II and synaptopodin in the same cells (Video S1).

Moreover, the in vivo *wt* mice at age 22 months, exposed to hypoxia (8% O_2_) for 24 h, revealed enhanced Arg-II levels in the whole kidney lysates (Figure 7A). Immunofluorescence staining demonstrated elevated Arg-II levels in the proximal tubules, as confirmed by co-localization with proximal tubular marker ACE1 (Figure 7B and Appendix A) as well as in the glomeruli (Figure 7C), which is in line with the results from ex vivo experiments with isolated kidneys. Moreover, the old *wt* mice exposed to hypoxia showed significantly decreased podocyte marker synaptopodin levels in the glomeruli (Figure 7D), which was prevented in *arg-ii^−/−^* mice (Figure 7D). In addition, albuminuria was reduced in the old *arg-ii^−/−^* mice as compared to the age-matched *wt* mice (Figure 8). The age-associated increase in albuminuria was associated with an increase in Arg-II staining in glomeruli, which was at least partly in podocytes, as shown by co-localization with synaptopodin (Appendix A). Hypoxia enhances albuminuria in the *wt* mice, which is, however, not significantly prevented in *arg-ii^−/−^* mice (Figure 8).

## 4. Discussion

Our present study reports an important role of Arg-II in renal podocyte injury induced by hypoxia, which is mediated by an enhanced mtROS resulting from elevated Arg-II levels in the cells.

The clinical and experimental studies demonstrate that podocyte injury importantly contributes to the development of progressive kidney disease [34]. Among the insults, hypoxia is well known to cause podocyte injury, contributing to the disruption of the integrity of glomerulus functions [10,35,36]. Aside from several pathological conditions, such as ischemia and type 2 diabetes, populations living at high altitudes reveal a higher incidence of end-stage renal disease [37,38,39], the so-called high-altitude renal syndrome [40]. The detrimental effects of hypoxia on podocytes include a decreased expression of proteins, which are important for the constitution of the foot process and slit diaphragm, resulting in foot-process effacement, slit-diaphragm disruption, and proteinuria [2]. It is well recognized that a normal podocyte shape and architecture are maintained by an intact cytoskeleton arrangement, which is critical for the physiological renal permselectivity and ultra-filtration of urine through the glomerular filtration barrier [41]. In line with these findings in the literature, our results show that hypoxia causes a derangement of cytoskeletal actin fibers, a decrease in podocin in cultured podocytes, and a decrease in synaptopodin—the proline-rich actin-associated protein—in kidneys exposed to hypoxic conditions in ex vivo and in vivo animal models. Importantly, these effects of hypoxia are accompanied by increased protein levels of Arg-II, HIF1α, and HIF2α, suggesting that HIFs and Arg-II may play a role in podocyte injury under hypoxic conditions. Indeed, silencing the *arg-ii* gene in podocytes reduces cytoskeleton actin fiber derangement and prevents a decrease in the podocin levels caused by hypoxia. Furthermore, the decrease in synaptopodin levels in isolated kidneys exposed to hypoxia or kidneys from mice exposed to hypoxia (8% O_2_ for 24 h) is ameliorated in *arg-ii^−/−^* mice. These results demonstrate that Arg-II promotes hypoxia-induced podocyte injury.

It is of note, however, that Arg-II is abundantly and constitutively expressed in proximal tubular cells (PTCs) and is highly inducible under various pathological and physiological conditions, such as type 2 diabetes, ischemia and/or hypoxia, as well as in renal aging [19,24,25]. In contrast to the PTCs, no study has shown Arg-II expression in podocytes until Li and colleagues most recently reported a role of Arg-II in mitochondrial dysfunction/remodeling in diabetic kidney disease [27]. They demonstrated that Tug1/PGC1α-mediated renoprotection in genetic type 2 diabetic mice is attributable to Arg-II suppression in podocytes. In line with this study, we showed that Arg-II is expressed in podocytes at low levels under normoxic conditions but elevated under hypoxic conditions, as demonstrated by the co-staining of Arg-II and synaptopodin. Of note, Arg-II staining is also found in glomerulus cells that are negative for synaptopodin, suggesting that other cells in the glomeruli, such as the endothelial cells and/or mesangial cells, could express Arg-II under hypoxic conditions. A detailed characterization of these cell types expressing Arg-II, and the functions of Arg-II in these glomerulus cells, require further investigation.

The mechanisms of Arg-II upregulation under hypoxic conditions have been reported to be mediated by HIFs in various cell types, including vascular cells and kidney PTCs [42,43,44,45,46,47]. Similar to the renal epithelial cells, Arg-II is upregulated under hypoxic conditions, as demonstrated in the present study in a cultured podocyte cell model, in ex vivo and in vivo animal models exposed to hypoxia. The upregulation of Arg-II is significantly inhibited by silencing *hif1α* but not by silencing *hif2α*, suggesting that HIF1α is the main driver of hypoxia-induced Arg-II upregulation in podocytes. These findings are similar to those reported in human vascular endothelial cells, in which hypoxia-induced upregulation of Arg-II is mediated through HIF1α [42]. The role of HIFs in upregulating Arg-II is further supported by the fact that the stabilization of HIFs by the PHD inhibitor DMOG mimicked the effects of hypoxia. Of note, although silencing HIF1α but not HIF2α is able to inhibit Arg-II upregulation, demonstrating the role of HIF1α in the upregulation of Arg-II, Arg-II upregulation under hypoxic conditions, however, seems more correlated with HIF2α rather than with the HIF1α levels. The explanation could be that the initial upregulation of HIF1α already triggers the downstream mechanism(s) for Arg-II expression, which does not necessarily require parallel sustained high levels of HIF1α, and only a lower but significant elevation of HIF1α in the later phase is sufficient to maintain Arg-II at high levels. We also noticed that silencing *hif1α* partially reduced the HIF2α levels in the podocytes, which is surprising. The specificity of siRNA for human *hif1α* has been demonstrated in our previous studies in human HK2 renal epithelial cells and endothelial cells [24,42]. It is most likely that there is a cell-specific interaction between HIF1α and HIF2α in the podocytes but not in HK2 and the endothelial cells. The underlying mechanism of the interaction requires further investigation. Although the functions of HIFs in renal diseases remain controversial, both the beneficial and detrimental effects of HIF inhibition on renal damage have been reported in the literature [48,49,50,51,52]. The results of our study suggest that targeting Arg-II would achieve more specific effects in protection against kidney damage since Arg-II lies downstream of HIFs, and multiple functions and downstream targets of HIFs exist [53].

Since Arg-II is a mitochondrial enzyme [20,21,22] and overexpression of Arg-II in the vascular cells has been reported to cause mitochondrial dysfunction, such as mtROS generation [42], we further investigated the hypothesis of whether Arg-II promotes mtROS, resulting in podocyte injury. Hypoxia is known to cause mitochondrial dysfunction and enhances mtROS generation and cellular damage [54]. Importantly, mtROS generation is induced in podocytes under hypoxic conditions and upon treatment with the HIFα stabilizing compound, DMOG. The enhanced mtROS generation and cell cytoskeleton fiber derangement under hypoxic conditions are prevented by silencing *arg-ii* and by treating the cells with the mitochondrial complex-I inhibitor, rotenone. In contrast to the wild-type control cells, in *arg-ii^−/−^* podocytes, neither hypoxia nor rotenone has a significant effect on the cytoskeleton fiber alterations, confirming the role of mtROS in Arg-II’s effect on podocyte damage. Moreover, a decrease in synaptopodin in the wild-type podocytes can be prevented by rotenone, demonstrating that mtROS generation is induced by Arg-II under hypoxic conditions and causes podocyte injury.

It is important to note, however, that *arg-ii^−/−^* mice exhibit reduced albuminuria at old age; however, they do not show protection against a hypoxia-induced increase in albuminuria as compared to age-matched *wt* mice. The results demonstrate a protective role of *arg-ii* deficiency in proteinuria in aging but not in hypoxic animal models, suggesting that *arg-ii* deficiency is sufficient to protect against age-associated proteinuria but not sufficient to protect mice in vivo against a hypoxia-induced increase in proteinuria, probably due to the Arg-II-independent mechanisms involved. The protective effect of *arg-ii*^−/−^ on albuminuria in aging is an interesting and relevant observation that needs further detailed investigation. Our recent study demonstrates an age-associated increase in the Arg-II levels in proximal tubular cells (19). In the present study, we also showed an age-associated increase in glomerulus Arg-II expression. These results could explain the age-associated increase in albuminuria, which is reduced in age-matched *arg-ii*^−/−^ mice. Since Arg-II is also found in the synaptopodin-negative cells in the glomeruli, future studies shall systematically investigate whether there is an age-associated increase in Arg-II in the mesangial cells and/or endothelial cells and whether the cell-specific Arg-II participates in the development of glomerulopathy and/or proteinuria associated with aging.

### Study Limitations and Perspectives

There are several limitations to our study. (1) To confirm the function of Arg-II in podocytes, experiments with podocyte-specific *arg-ii*^−/−^ mouse models shall be developed in the future. (2) Another aspect is that systemic hypoxia affects many organ functions, which will profoundly influence the impact of *arg-ii^−/−^* in the kidney. A renal ischemia model could be considered in the future. (3) GFR is not analyzed, which requires a precise and reliable measurement in the mouse models. (4) Although the results of our experiments with cell culture models, isolated kidneys, and mouse models strongly suggest a role of Arg-II in the podocyte injury/dysfunction associated with hypoxia and aging, which is linked to mitochondrial oxidative stress, the detailed molecular mechanisms of Arg-II-induced podocyte injury and/or podocyte aging remain to be investigated. (5) As discussed above, the questions of whether Arg-II is also induced or expressed in the mesangial and glomerulus endothelial cells and how these cells, if Arg-II expression levels are increased, could influence podocyte function remains unknown.

Despite these limitations, our study explored a novel role of Arg-II in podocyte dysfunction and demonstrated that Arg-II upregulation causes mitochondrial ROS generation to participate in podocyte injury. Although *Arg-ii* ablation seems insufficient to protect mice in vivo against a hypoxia-induced increase in albuminuria, it is able to reduce albuminuria in aging. This finding may pave a new avenue for future research into renal podocyte biology in the pathophysiology of kidney disease.

## Figures and Tables

**Figure 1 biomolecules-12-01213-f001:**
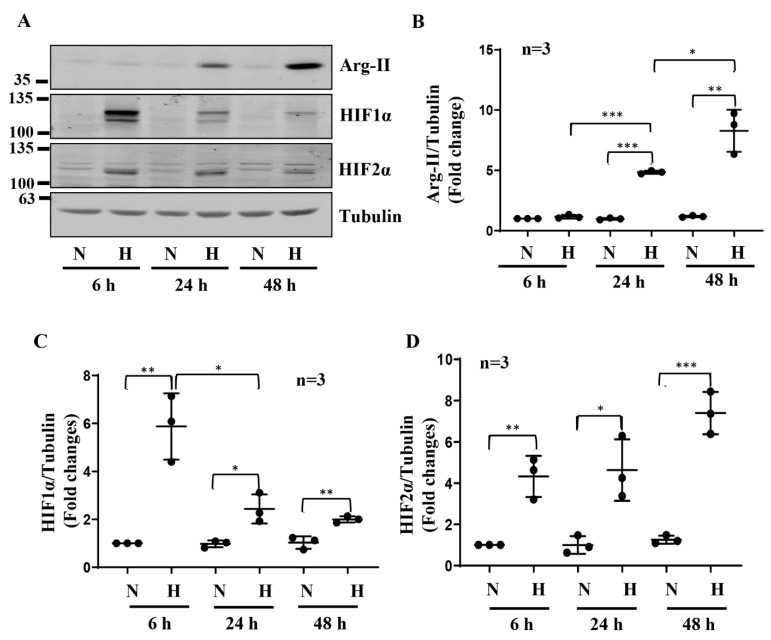
Hypoxia enhances Arg-II levels and HIFs in human podocytes. (**A**) Representative immunoblotting results of the protein levels of Arg-II, HIF1α, and HIF2α in different groups of podocytes. Tubulin serves as the loading control. (**B**–**D**) Quantification of the signals of Arg-II, HIF1α, and HIF2α, respectively. * *p* < 0.05, ** *p* < 0.01, *** *p* < 0.001 between the indicated groups. n = 3; N: normoxia; H: hypoxia, h = hours.

**Figure 2 biomolecules-12-01213-f002:**
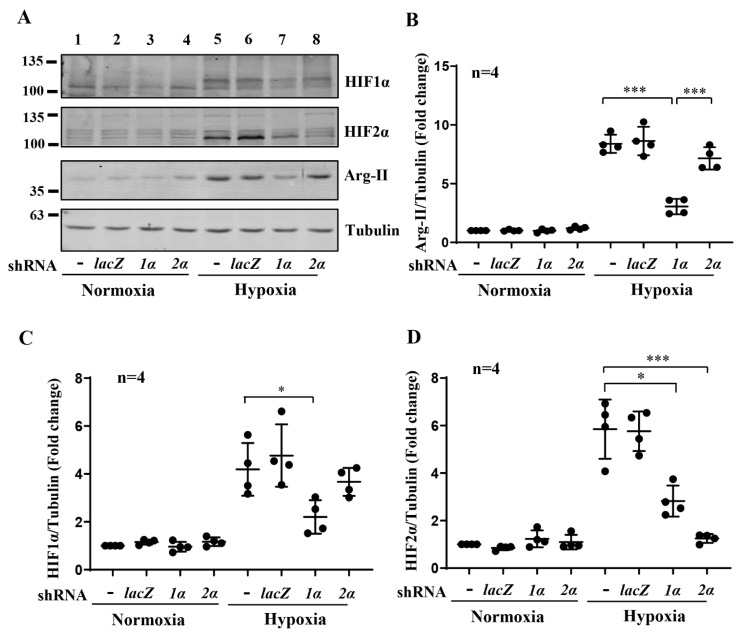
Role of HIFs in Arg-II upregulation by hypoxia in human podocytes. (**A**) Representative immunoblotting results of the protein levels of Arg-II, HIF1α, and HIF2α in different groups of podocytes. Tubulin serves as the loading control. (**B–D**) Quantification of the signals of Arg-II, HIF1α, and HIF2α, respectively. Lanes 1 and 5: wild-type podocytes AB8/13; Lanes 2 and 6: podocytes transduced with control rAd/U6-*lacZ*^shRNA^; Lanes 3 and 7: podocytes transduced with rAd/U6-*hif1**α*^shRNA^ (*1**α*); Lanes 4 and 8: podocytes transduced with rAd/U6-*hif2**α*^shRNA^ (*2**^α^*). * *p* < 0.05, *** *p* < 0.001 between the indicated groups. n = 4.

**Figure 3 biomolecules-12-01213-f003:**
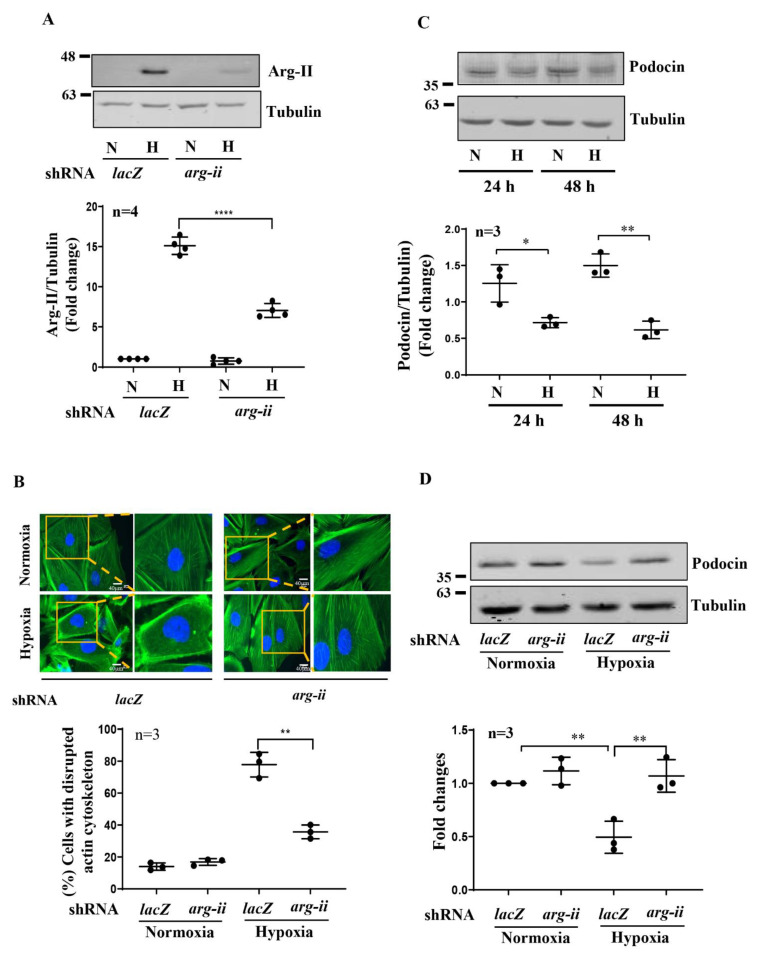
Silencing *arg-ii* in human podocytes attenuates hypoxia-induced cytoskeleton actin fiber derangement. (**A**) Representative immunoblotting results of Arg-II protein levels in different groups of podocytes. Tubulin serves as the loading control; n = 4. (**B**) Representative images showing phalloidin staining of cytoskeleton actin fibers in different groups of podocytes. Nucleoli were stained with DAPI (blue). Inserts illustrate a cell that represents the typical changes in cytoskeletal fibers under the indicated conditions. The graphics below show the quantification of podocytes with a disrupted cytoskeleton. n = 3. (**C**) Representative immunoblotting results show decreased podocin protein levels under hypoxic conditions; Tubulin serves as the loading control. The graphics below show the quantification of podocin signals. (**D**) Representative immunoblotting results show protective effects of *arg-ii* silencing on a hypoxia-induced decrease in podocin protein levels in podocytes. Tubulin serves as the loading control. The graphics below show the quantification of podocin signals. N: normoxia; H: hypoxia. * *p* < 0.05, ** *p* < 0.01, **** *p* < 0.001 between the indicated groups. n = 3.

**Figure 4 biomolecules-12-01213-f004:**
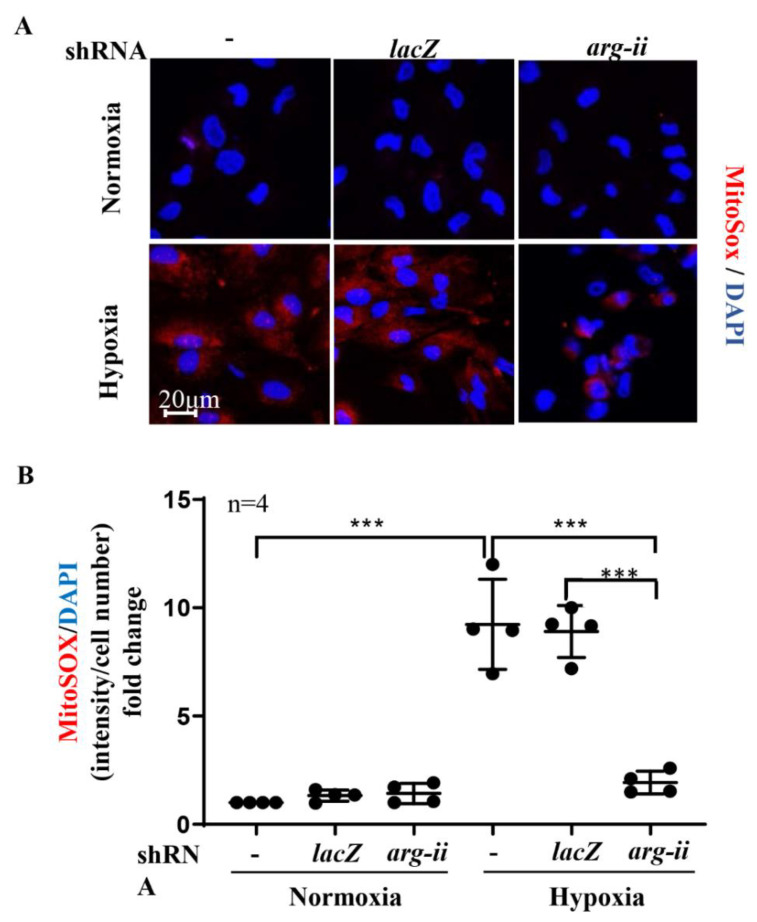
Silencing *arg-ii* reduces hypoxia-enhanced mitochondrial ROS production in podocytes. MitoSOX Red reagent staining was used to analyze mitochondrial ROS production. -: wild-type podocytes; shRNA-*lacZ*: podocytes transduced with rAd/U6-*lacZ*^shRNA^ as controls; shRNA-*arg-ii*: podocytes transduced with rAd/U6-*arg-ii*^shRNA^. (**A**). Representative images of MitoSOX staining; (**B**). Quantification of relative fluorescence fold-change in different groups of podocytes. *** *p* < 0.001 between the indicated groups. n = 4.

**Figure 5 biomolecules-12-01213-f005:**
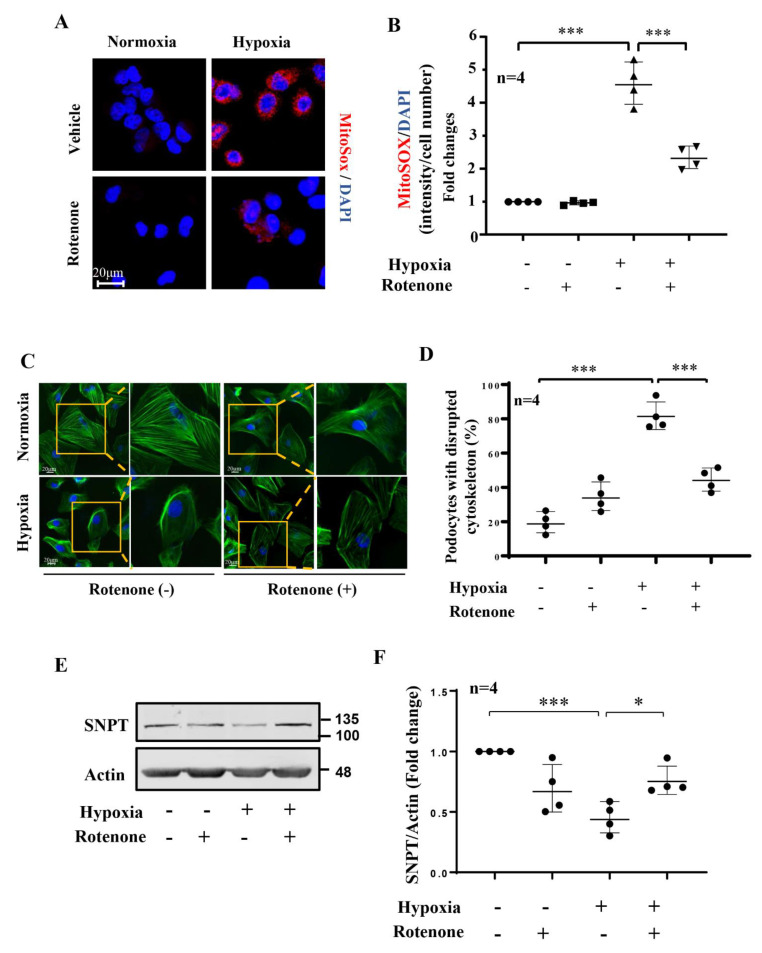
Role of mitochondrial ROS on hypoxia-induced podocyte injury. The cells were pretreated with or without rotenone (2 μmol/L) for 1 h and then incubated under normoxia or hypoxia (2% O_2_) conditions for 24 h. (**A**). Representative images show MitoSOX staining in human podocytes under different conditions. Nucleoli were stained with DAPI (blue). (**B**). Quantification of relative fluorescence fold-change in different groups of podocytes. (**C**). Representative images show phalloidin staining of cytoskeletal actin fibers in human podocytes under different conditions. Nucleoli were stained with DAPI (blue). Inserts illustrate a cell that represents the typical changes in cytoskeletal fibers under the indicated conditions. (**D**). Quantification of the podocytes with the disrupted actin cytoskeleton. (**E**). Representative immunoblotting results of the protein levels of synaptopodin (SNPT) in different groups of podocytes; β-actin serves as the loading control. (**F**). Quantification of synaptopodin signals. * *p* < 0.05, *** *p* < 0.001 between the indicated groups. n = 4.

**Figure 6 biomolecules-12-01213-f006:**
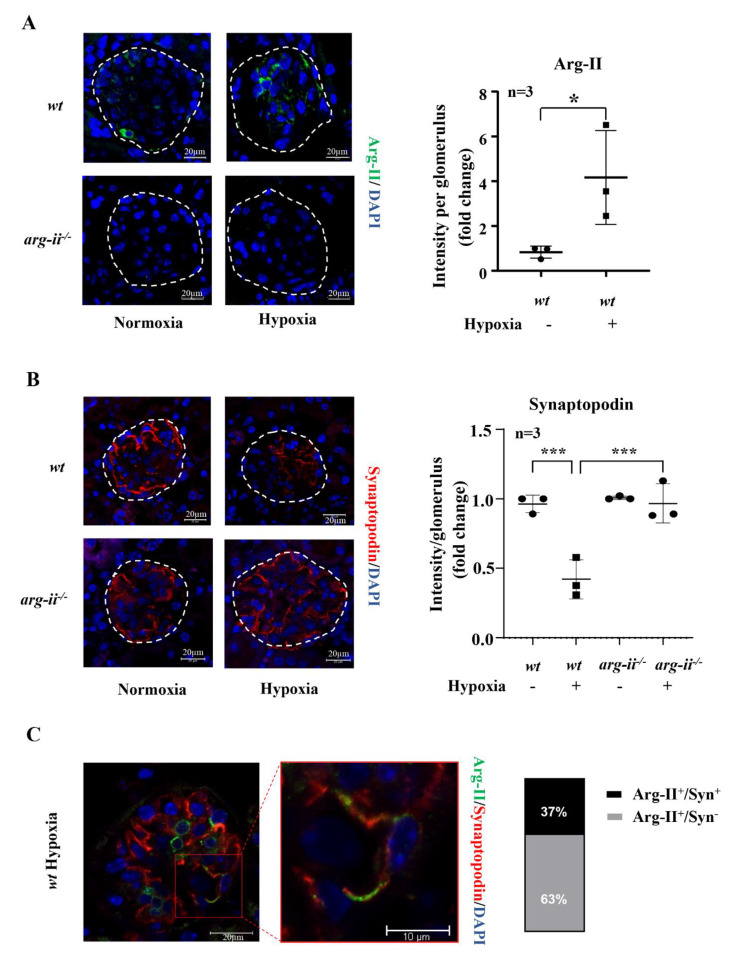
*Arg-ii* ablation prevents hypoxia-induced synaptopodin reduction in glomeruli ex vivo. Confocal immunofluorescence staining of (**A**) Arg-II, (**B**) synaptopodin and (**C**) co-localization of Arg-II (green) and synaptopodin (red) with quantification of Arg-II^+^/synaptopodin^+^ and Arg-II^+^/synaptopodin^−^ signals in isolated kidneys from *wt* and *arg-ii*^−/−^ mice that were exposed to normoxia or hypoxia conditions (1% O_2_, 24 h) ex vivo. Nucleoli were stained with DAPI (blue). n = 3, * *p* < 0.05, *** *p* < 0.001 between the indicated groups.

**Figure 7 biomolecules-12-01213-f007:**
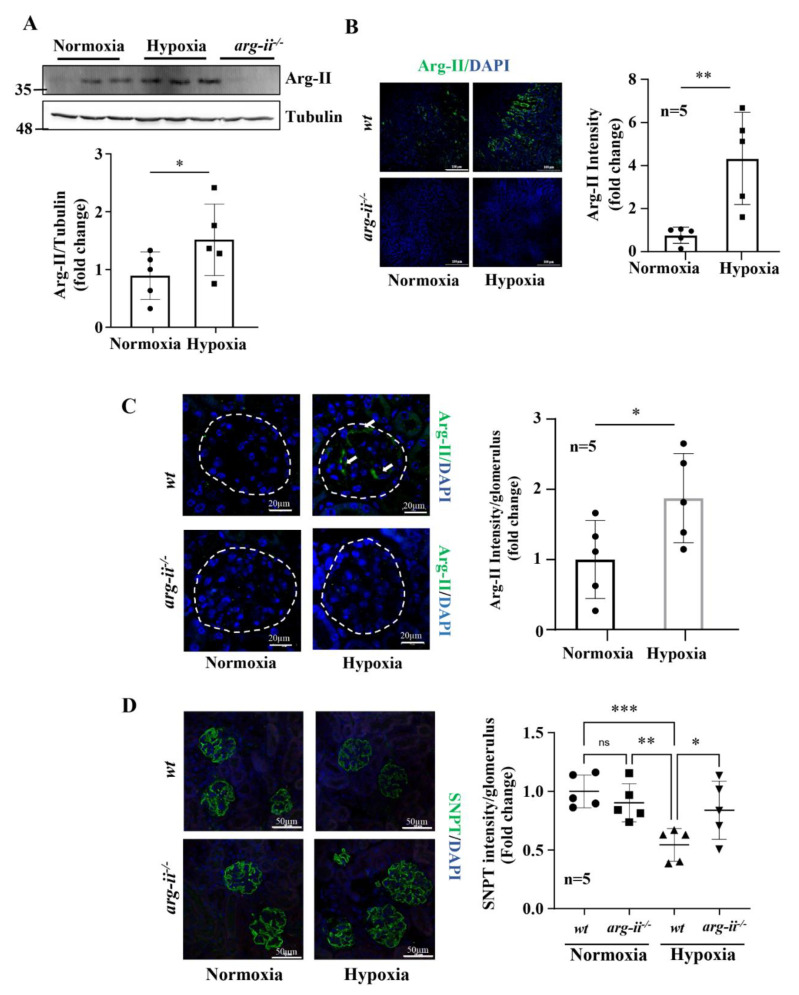
*Arg-ii* ablation prevents hypoxia-induced reduction in synaptopodin in vivo. (**A**). Representative immunoblotting shows Arg-II levels and relative quantification of Arg-II levels in *wt* under normoxic and hypoxic conditions. Tubulin serves as the loading control. (**B**) Representative images show Arg-II protein expression and quantification in proximal tubular cells (PTCs). (**C**) Confocal immunofluorescence staining shows Arg-II protein expression and quantification in glomeruli; (**D**). Confocal immunofluorescence staining of synaptopodin (SNPT) and quantification in *wt* and *arg-ii^−/−^* mouse kidneys (n = 5 for each group). Nucleoli were stained with DAPI (blue). * *p* < 0.05, ** *p* < 0.01, *** *p* < 0.001 between the indicated groups.

**Figure 8 biomolecules-12-01213-f008:**
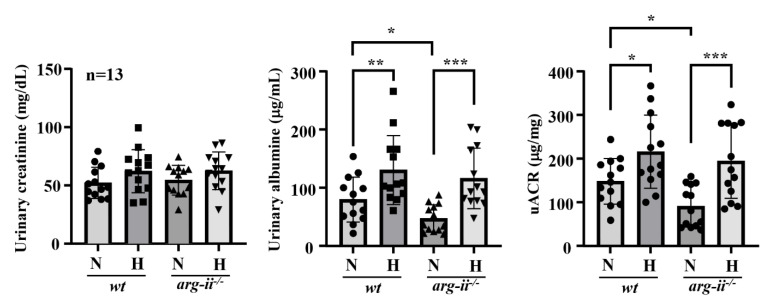
Effect of *arg-ii* ablation on albuminuria. Urinary creatinine and albuminuria were assessed in 20 to 24 months old mice exposed to normoxia controls or hypoxia (8% O_2_ for 24 h). The urinary albumin/creatinine ratio (uACR) was evaluated in *wt* and *arg-ii^−/−^* animals. n = 13 in each group. * *p* < 0.05, ** *p* < 0.01, *** *p* < 0.005 between the indicated groups.

## Data Availability

The datasets used and/or analyzed during the current study are available from the corresponding author upon reasonable request.

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
