# Peer review of "Role of Arginase-II in Podocyte Injury under Hypoxic Conditions"

_biomolecules, 2022, doi:10.3390/biom12091213_

Round 1
Reviewer 1 Report
In the paper “Role of arginase-II in podocyte injury under hypoxic conditions” the authors investigated the possible effects of hypoxia on Arginase II in podocytes, using human cell lines and control or arg-II deficient mice. They found that hypoxia causes a time-dependent upregulation of Arg-II, an effect mediated by the stabilization of HIF1A factor. Using a combination of in vitro and ex vivo approaches, they found that arginase II is directly involved in podocytes injury and its upregulation correlates with increased mitochondrial ROS production - derangement of cellular actin cytoskeletal fibres- and downregulation of synaptopodin levels. As such, the effects of hypoxia on podocytes injury can be mimicked by HIF1A stabilization via DMGO preincubation and rescued by Arg II silencing. The results of this study provide important insight in podocyte biology under pathophysiological conditions.
Overall, the findings are novel and important. The experiments are generally well designed and follow a logical order to support the authors' conclusion. The authors should consider the following issues:
Figure 1-2 and S1. The siRNA experiments clearly showed that hypoxia-induced ArgII upregulation is mediated by HIF1a, but not HIF2a. However, counterintuitively, the hypoxia induced time-dependent upregulation of ArgII correlated positively with HIF2a, but not with HIF1a (fig1A). Similarly, in supplementary fig 1, DMOG treatment (PHD1 inhibitor) is expected to stabilize, and therby increase, HIF levels. In line with this hypothesis, there is a dose-dependent upregulation of HIF2 and ArgII, but not HIF1A, which is even downregulated. Could you please explain?
Figure 6 panel C. The colocalization image shows that only a minority of synaptopodin-positive cells are also positive for ArgII and that ArgII is mainly expressed by synaptopodin-negative cells (a quantification analysis would clarify this point). If ArgII is expressed in mesangial and/or endothelial cells (as hypothesized by the author) how it is expected to impact on podocytes function?
Fig.8 Why, under normoxic conditions, is there a statistically significant decrease in urinary albumin and uACR in ArgII ko mice compared to age-matched littermate? Is there an age-dependent induction of ArgII during ageing that could account for this effect? Please, comment in the text.
Minor
3B, 5C High magnification images (also one cell per image) should be inserted to appreciate changes in the cytoskeleton organization induced by the different treatments
Author Response
In the paper “Role of arginase-II in podocyte injury under hypoxic conditions” the authors investigated the possible effects of hypoxia on Arginase II in podocytes, using human cell lines and control or arg-II deficient mice. They found that hypoxia causes a time-dependent upregulation of Arg-II, an effect mediated by the stabilization of HIF1A factor. Using a combination of in vitro and ex vivo approaches, they found that arginase II is directly involved in podocytes injury and its upregulation correlates with increased mitochondrial ROS production - derangement of cellular actin cytoskeletal fibres- and downregulation of synaptopodin levels. As such, the effects of hypoxia on podocytes injury can be mimicked by HIF1A stabilization via DMGO preincubation and rescued by Arg II silencing. The results of this study provide important insight in podocyte biology under pathophysiological conditions.
Thank you very much for your positive opinion on our manuscript!
Overall, the findings are novel and important. The experiments are generally well designed and follow a logical order to support the authors' conclusion. The authors should consider the following issues:
Figure 1-2 and S1. The siRNA experiments clearly showed that hypoxia-induced ArgII upregulation is mediated by HIF1a, but not HIF2a. However, counterintuitively, the hypoxia induced time-dependent upregulation of ArgII correlated positively with HIF2a, but not with HIF1a (fig1A). Similarly, in supplementary fig 1, DMOG treatment (PHD1 inhibitor) is expected to stabilize, and thereby increase, HIF levels. In line with this hypothesis, there is a dose-dependent upregulation of HIF2 and ArgII, but not HIF1A, which is even downregulated. Could you please explain?
Thank you very much for raising this point! Indeed, silencing HIF1α but not HIF2α is able to inhibit Arg-II upregulation, demonstrating the role of HIF1α in upregulation of Arg-II. However, Arg-II upregulation under hypoxic conditions seems more correlated with HIF2α rather than with HIF1α levels. The explanation could be that the initial upregulation of HIF1α already triggers down-stream mechanisms for Arg-II expression which does not necessarily require parallel sustained high levels of HIF1α; only a lower but significant elevation of HIF1α in the later phase is sufficient to maintain Arg-II at high levels, which is the case for transient activation of many signaling pathways regulating gene expression and cellular responses. We have added this discussion in the Discussion section, the 4th paragraph.
Figure 6 panel C. The colocalization image shows that only a minority of synaptopodin-positive cells are also positive for ArgII and that ArgII is mainly expressed by synaptopodin-negative cells (a quantification analysis would clarify this point). If ArgII is expressed in mesangial and/or endothelial cells (as hypothesized by the author) how it is expected to impact on podocytes function?
According to this reviewer’s comment, quantification of synaptopodin-positive and negative cells that reveal Arg-II expression was performed. The results show that 37% of Arg-II signals are co-localized with synaptopodin, while 63% of Arg-II are not co-localized with synaptopodin, suggesting that Arg-II is also significantly induced under hypoxic conditions in glomerulus mesangial cells and/or endothelial cells. Confocal Z-stack recording confirms the localization of Arg-II and synaptopodin in the same cells (Please see the Suppl. video file). We have added the quantitative information on Arg-II/synaptopodin co-localization in the Fig. 6C. The results are described in the result section point 6, 1st paragraph. The legend to Fig. 6C has been adapted accordingly. The impact of Arg-II in mesangial cells and endothelial cells on podocyte function remains an interesting research topic for further research as we have discussed in the manuscript (Discussion section, the last 2nd and 3rd paragraphs). We believe that this aspect is beyond the focus of this manuscript and did not go into detailed speculation.
Fig. 8 Why, under normoxic conditions, is there a statistically significant decrease in urinary albumin and uACR in ArgII ko mice compared to age-matched littermate? Is there an age-dependent induction of ArgII during ageing that could account for this effect? Please, comment in the text.
This is a very interesting and relevant question. Our recent study demonstrates an age-associated increase in Arg-II levels in proximal tubular cells (19). We performed new experiments with confocal microscopy and demonstrated that indeed glomerulus Arg-II signals are higher in aged mice as compared with the young animals (Please see Suppl. Fig. 7). The results may explain the age associated increase in albuminuria which is reduced in age-matched arg-ii-/- mice. We have discussed this aspect in the Discussion Section, the last 2nd and 3rd paragraphs).
Minor
3B, 5C High magnification images (also one cell per image) should be inserted to appreciate changes in the cytoskeleton organization induced by the different treatments.
Inserts with one cell per image were presented to appreciate changes in cytoskeleton organization as suggested by this reviewer in Fig. 3B and 5C. Legends to these two figures were adapted.
We sincerely hope that our revised manuscript is now suitable for publication!
Reviewer 2 Report
The study by Ren et al is comprehensive and relevant. The introduction sets the scene by describing role of hypoxia in acute and chronic renal injury. The methodology is clear and concise and provide necessary information to repeat the method as well as critical steps. The figures outline the key message and results are well presented. However, there are some points need to be addressed in the Discussion part:
1. How do you explain the result that silencing hif1α is accompanied with partial inhibition of hypoxia-induced HIF2a accumulation?
2. How do you explain that albuminuria is reduced in the old arg-ii-/- mice as compared to the age-matched wt mice?
3. Limitation of the study
4. The conclusions needs to be highlighted.
Author Response
The study by Ren et al is comprehensive and relevant. The introduction sets the scene by describing role of hypoxia in acute and chronic renal injury. The methodology is clear and concise and provide necessary information to repeat the method as well as critical steps. The figures outline the key message and results are well presented. However, there are some points need to be addressed in the Discussion part:
Thank you very much for your positive opinion on our manuscript!
- How do you explain the result that silencing hif1α is accompanied with partial inhibition of hypoxia-induced HIF2a accumulation?
Thanks for your critical comment on this result! It is true that silencing hif1α partially reduced HIF2α levels in the podocytes, which remains unexplained. The specificity of siRNA for human hif1α has been demonstrated in our previous studies in human HK2 renal epithelial cells and endothelial cells [24, 36]. It is most likely that there is a cell specific interaction between HIF1α and HIF2α in the podocytes but not in HK2 and endothelial cells. The underlying mechanism of the interaction of HIFs in podocytes requires further investigation. We have discussed this point in the Discussion Section, the 4th paragraph.
- How do you explain that albuminuria is reduced in the old arg-ii-/- mice as compared to the age-matched wt mice?
This is a very interesting and relevant question. Our recent study demonstrates an age-associated increase in Arg-II levels in proximal tubular cells (19). We performed new experiments with confocal microscopy and demonstrated that indeed glomerulus Arg-II signals are higher in aged mice as compared with the young animals (Please see Suppl. Fig. 6). The results may explain the age associated increase in albuminuria which is reduced in age-matched arg-ii-/- mice. We have discussed this aspect in the Discussion Section, the last 2nd and 3rd paragraphs).
- Limitation of the study
We have discussed several limitations of our study in the Discussion Section, the 2nd last paragraph.
- The conclusions needs to be highlighted.
Yes, we have highlighted the conclusion according to this reviewer’s suggestion in the Discussion Section, the last paragraph.
We sincerely hope that our revised manuscript is now suitable for publication!
Reviewer 3 Report
The manuscript by Ren et al entitled “Role of arginase-II in podocyte injury under hypoxic conditions” evaluated the role of Arg-II in hypoxia-induced podocyte injury through HIF-1α modulation, podocyte cytoskeleton maintenance, and ROS production by mitochondria. The study is interesting and contributes to knowledge in the field. Please acknowledge the following comments before proceeding.
Major comments
1) To further substantiate the currents findings and provide mechanistic insights, did the authors verify the use of the mitochondrial complex-I inhibitor rotenone in shRNA-Arg-ii podocytes? Please provide the results of this experiment under hypoxic conditions (immunofluorescence and cytoskeleton analyses).
2) In Figure 6C, please provide confocal images of Z-stack to demonstrate the co-localization of Arg-II and synaptopodin.
3) Likewise, was co-localization of Arg II found with proximal tubules? Was Arg-II equally found in the medullary compartment?
4) In Figure 8, why a P-value ≤P.05 was considered significant instead of a P-Value <0.05, as documented for other experiments throughout the manuscript? Likewise, correct the P-value in the Methods section (line 427).
5) As higher levels of proteinuria were not prevented in Arg-ii -/- knockout mice, did the author evaluate the glomerular filtration rate?
Minor comments
1) In the methods section, describe how many replicates were used for the in vitro experiments. Likewise, include the list of reagents in each section that they were used.
2) In the legends of Figures 3C and 3D, describe that podocin levels were evaluated without and with silencing Arg-II, respectively, to facilitate the understanding of the experiment by readers.
3) Indicate the scale bars of Figures 7B and 7D.
4) Standardize the description of the P-value throughout the manuscript. Sometimes it appears in italics, other times without italics; sometimes it appears with a capital letter, other times with a lower case.
Author Response
Responses to Reviewer Nr. 3:
The manuscript by Ren et al entitled “Role of arginase-II in podocyte injury under hypoxic conditions” evaluated the role of Arg-II in hypoxia-induced podocyte injury through HIF-1α modulation, podocyte cytoskeleton maintenance, and ROS production by mitochondria. The study is interesting and contributes to knowledge in the field. Please acknowledge the following comments before proceeding.
Thank you very much for your positive opinion on our manuscript!
Major comments
- To further substantiate the currents findings and provide mechanistic insights, did the authors verify the use of the mitochondrial complex-I inhibitor rotenone in shRNA-Arg-ii podocytes? Please provide the results of this experiment under hypoxic conditions (immunofluorescence and cytoskeleton analyses).
According to this reviewer’s suggestion, we performed experiments in arg-ii knockout podocytes generated by CRISPR/Cas9 technology (because siRNA silencing can not fully eliminate Arg-II expression levels). The arg-ii knockout was confirmed by immunoblotting shown in Suppl. Fig. 5A. We found that in contrast to wild type cells (as shown in Fig. 5C), neither hypoxia nor rotenone has significant effect on cytoskeleton fiber alterations (Suppl. Fig. 5B), demonstrating that Arg-II exerts it’s effect through mtROS. These new results are included in the result section: “5. Role of mtROS in hypoxia-induced podocyte injury” and discussed in the discussion section, the 5th paragraph. The CRISPR/Cas9 method generating arg-ii-/- cell line was added to the Method Section: “Generation of arg-ii knockout cell line using CRISP/Cas9 technologies. The legend to the new Suppl. Fig. 5 was added to the file: “Suppl. figure legends”. The original uncropped blot was provided.
2) In Figure 6C, please provide confocal images of Z-stack to demonstrate the co-localization of Arg-II and synaptopodin.
As suggested by this reviewer, we performed new experiments for Z-stack recording and confirmed the localization of Arg-II and synaptopodin in the same cells (Please see the Suppl. video file). The result is described in the Result Section, point 6, 1st paragraph.
3) Likewise, was co-localization of Arg II found with proximal tubules. Was Arg-II equally found in the medullary compartment?
As suggested by this reviewer, we performed new experiments showing that under hypoxic conditions, Arg-II is expressed in proximal tubular cells as demonstrated by co-localisation with proximal tubular marker ACE1 (Suppl. Fig. 7) which confirmed our previous finding (19). This new data was described in the Result Section, the last paragraph. Arg-II was not found in the medullary compartment under this condition (data not shown in the manuscript). The anti-ACE1 antibody is now added to the Reagent paragraph of Method Section and also listed in the Suppl. table 1.
4) In Figure 8, why a P-value ≤P.05 was considered significant instead of a P-Value <0.05, as documented for other experiments throughout the manuscript? Likewise, correct the P-value in the Methods section (line 427).
Thank you very much for detecting the inconsistency! These have been corrected.
5) As higher levels of proteinuria were not prevented in Arg-ii-/- knockout mice, did the author evaluate the glomerular filtration rate?
Unfortunately, GFR was not analyzed. A precise and reliable method measuring GFR in mice are not available in the lab. We have discussed this point in the limitation of the study (please see the 2nd last paragraph).
Minor comments
- In the methods section, describe how many replicates were used for the in vitro experiments. Likewise, include the list of reagents in each section that they were used.
The experiments were repeated independently several times. Replication or number of replicated experiments are indicated in each figure with n and also described in legends to figures. Because the replicates are not always the same for all the sets of experiments, we prefer to indicate n numbers in each individual figure instead in the method in general. As standard, we prefer to have our reagents listed in the 1st paragraph of Materials and Methods.
- In the legends of Figures 3C and 3D, describe that podocin levels were evaluated without and with silencing Arg-II, respectively, to facilitate the understanding of the experiment by readers.
This has been added accordingly.
3) Indicate the scale bars of Figures 7B and 7D.
Thank you very much! The scale bars are added.
4) Standardize the description of the P-value throughout the manuscript. Sometimes it appears in italics, other times without italics; sometimes it appears with a capital letter, other times with a lower case.
Thank you very much for pointing out this inconsistency! We have standardized the description as p < …throughout the manuscript.
Round 2
Reviewer 3 Report
The authors addressed all comments. Congratulations on your work.
Author Response
Thanks.